# Reimplantation after Periprosthetic Joint Infection: The Role of Microbiology

**DOI:** 10.3390/antibiotics11101408

**Published:** 2022-10-13

**Authors:** Virginia Suardi, Nicola Logoluso, Filippo Maria Anghilieri, Giuseppe Santoro, Antonio Virgilio Pellegrini

**Affiliations:** 1Department of Reconstructive Surgery of Osteo-Articular Infections, IRCCS Istituto Ortopedico Galeazzi, 20100 Milan, Italy; 2Resident in Orthopedia and Traumatology, University of Milan, 20100 Milan, Italy

**Keywords:** periprosthetic joint infection (PJI), microbiology, two-stage revision, cultural test, reimplantation

## Abstract

Periprosthetic joint infection (PJI) is among the most feared orthopedic complications. Critical questions are whether the infection is completely resolved before reimplantation and what the clinical significance of positive culture is at reimplantation. The aim of this study was to determine whether a correlation exits between culture results at reimplantation after spacer insertion for hip and knee PJI and treatment failure rate. The data of 84 patients who underwent two-stage exchange arthroplasty for hip or knee PJI were reviewed and the results of intraoperative culture at reimplantation were analyzed quantitatively and qualitatively. Correlations were sought between these patterns and treatment outcome. Our data indicate no evidence for a correlation between positive culture at reimplantation and greater risk of treatment failure. Nonetheless, we noted a higher, albeit statistically not significant rate of treatment failure in patients with at least two samples testing positive. The role of microbiology at reimplantation remains unclear, but a positive culture might signal increased risk for subsequent implant failure. Further studies are needed to elucidate the implications of this finding.

## 1. Introduction

Periprosthetic joint infection (PJI) is among the most feared orthopedic complications. Arthroplasty infection is a leading cause of reduced quality of life compared to the general population [1]. The first step in the treatment of PJI is diagnosis by standardized definition according to scientific societies and organizations [2]. The second step is the development of more refined biomarkers and culture techniques to improve diagnostic skills [3]. After diagnosis, PJI is then treated by the debridement, antibiotics, and implant retention (DAIR) procedure and one-stage reimplantation or two-stage replacement arthroplasty. The pathogen, together with other known risk factors, plays a major role in the development and recurrence of PJIs. Despite advances in the diagnosis and treatment of arthroplasty infection, there is no gold standard diagnostic test or treatment [4].

Two-stage revision is the most common treatment yet despite accurate diagnosis and surgery failure rates of up to 40% are reported in some case series [5,6]. One of the reasons for poor outcome is the timing of second-stage surgery. The question is how to be sure that infection has been eradicated before reimplantation. A variety of diagnostic tests have been evaluated for their ability to exclude or confirm PJI but none to date have been sufficiently tested at the second stage of diagnosis and treatment.

Some studies have tested the validity of the Musculoskeletal Infection Society (MSIS) diagnostic criteria for PJI in patients with an antibiotic spacer just before reimplantation. The MSIS criteria have demonstrated good ability to confirm the persistence of infection (specificity > 89%, and so are considered good RULE IN) but cannot exclude infection because of their low sensitivity [7,8,9]. Erythrocyte sedimentation rate (ESR) and the C-reactive protein (CRP) assay are widely used, though their sensitivity is low when combined with an antibiotic-loaded spacer [10] and their levels can remain high long after recent surgery. 

Preoperative arthrocentesis is considered a mainstay in the diagnosis of PJI, but it may not be useful when an antibiotic-loaded spacer is placed. Studies have questioned its sensitivity in persistent infection, particularly infection caused by low-virulence pathogens [11,12]. The procedure is also difficult to perform with hip spacers present. The role of rapid testing of intraoperative joint fluid (e.g., leukocyte esterase strip test) holds promise for diagnosing persistent infection; however, studies in this regard are still few [8,9,12,13,14,15]. 

As there is no gold standard for establishing resolution of infection and correct timing of reimplantation, orthopedic surgeons need to rely on their experience and ability in interpreting what data are available. Evidence for the persistence of infection can be obtained from microbiological testing at reimplantation; however, culture results take time. The 2018 Philadelphia Consensus Conference recommends collecting four to six tissue samples for interoperative culture during reimplantation in both septic and aseptic revision. Although sampling is strongly recommended, it is not certain whether a positive finding necessarily means outcome failure. The aim of the present study is to report our experience about the hypothetical correlation between culture results at reimplantation after spacer insertion for treating hip and knee PJI and treatment failure rate.

## 2. Results

### 2.1. Patient Demographics

The study population was 84 patients (44 men and 40 women, median age 72 years, range 42–85). There were no differences between patients with and without reinfection when stratified by demographic (age, sex, BMI (body mass index, weight in kg divided by height in meters squared), tobacco use) and clinical characteristics (diabetes, fistula, days with spacer, duration of second-stage surgery), except for previous surgery (Table 1). Infection reoccurred only in patients *(n* = 4) with previous revision for infection. The difference in frequency of infection was statistically different from patients who did not undergo revision or who underwent aseptic revision (*p* = 0.003). Multivariate analysis showed no correlation between demographics (age, sex, tobacco use, BMI) or clinical variables (diabetes, fistula, previous surgery) and recurrence of infection. 

### 2.2. Microbiological Testing 

Microbiological testing showed positive results for 25/84 spacers. Table 2 lists the pathogens isolated at first- and second-stage surgery. Two or more samples were positive in 10 cases and only one sample was positive in 15. No further infections were observed in patients with one positive sample, while two failures were observed in patients with more than one positive sample. Treatment failure was recorded in two patients with negative samples but no failure in patients with a single positive sample; this difference was statistically significant (*p* = 0.047). A new germ was isolated in 18 cases at revision (one positive sample in 12 cases) and a persistent pathogen was isolated in 7 cases (3 of which with only one positive sample). In the two cases of failure, one was caused by a new pathogen and the other by persistent infection with the same pathogen. More than one germ (polymicrobial infection) was isolated in 19 cases at the first and in five at the second stage. 

Spacers may have been contaminated in cases with a single positive tissue sample. Two treatment failures were recorded in the 74 (2.7%) patients with a negative spacer and two failures in the 10 (20%) patients with a positive spacer; the difference was not statistically significant (*p* = 0.105).

### 2.3. Analysis of Results by Subgroup

The final study population was 84 cases of PJI (*n* = 60 knee, *n* = 24 hip). The medical charts were gleaned for patient age and sex, comorbidities, BMI, tobacco use and previous culture results. Multiple subgroups were formed according to culture results at reimplantation (Figure 1). Group A (*n* = 59, 70.2%) patients had a negative culture and group B (*n* = 25, 29.8%) had a positive culture; recurrence of infection occurred in 2 (3.4%) of those with a negative culture and in 2 (8%) of those with a positive culture. Group B was further subdivided in group B1 (*n* = 15/25, 60%), with no recurrence of infection, and group B2 (*n* = 10/25, 40%), with recurrence of infection in 2 (20%) patients. Patients with a positive culture were categorized according to the type of pathogen isolated. Groups were formed of patients with superinfection (group S) caused by a pathogen other than the one isolated at explantation: 12/15 patients, 1 of which with one positive sample (subgroup S1); no failures were recorded for this group; 6/10 patients had at least two positive samples (subgroup S2) and one (16.7%) failure was recorded for this subgroup. Another group (subgroup termed P for persistent germ) was formed of patients in which the same pathogen was isolated as that at explantation: no failure was noted for 3/15 patients with one positive culture (subgroup P1); one (20%) failure was recorded in 4/10 patients with at least two positive cultures (subgroup P2).

The patients were then stratified by outcome according to Delphi criteria (treatment success vs. failure) and the correlations between microbiological test results and clinical outcome were analyzed.

Analysis of our data shows that the failure rate was higher in patients with a positive than in those with a negative culture: 3.3% in group A, 8% in group B, and 20% in group B2.

## 3. Discussion

Since the published data are scarce and discordant, it is difficult to attribute clinical significance to a positive culture test at prosthesis reimplantation. In our series, 29.7% of cultures tested positive at reimplantation; this rate is consistent with previous reports of rates of up to 44% [16]. Caution is warranted when interpreting this finding, however, because one or more positive cultures at reimplantation are not necessarily a cause of treatment failure. Furthermore, a single positive culture at reimplantation may result from contamination rather than indicate a true infection. 

Some studies found no correlation between positive culture at reimplantation and treatment failure, whereas others reported an increased risk of infection recurrence. In their study involving 48 patients undergoing two-stage revision for PJI in total knee arthroplasty, Hart et al. found 11 (22.9%) positive cultures at reimplantation. The pathogens were different from those isolated at explantation in seven cases; no correlation was found between culture results and treatment outcome at a minimum follow-up of 26 months [17]. In a more recent study, Bejon et al. analyzed the data from 152 patients undergoing two-stage revision for PJI with a follow-up of 4 years. Microbiological tests were positive in 21 cases (14%): a pathogen different from the one isolated at explantation was identified in 10 (6%), the pathogens were the same in 4 (3%), and culture at explantation was positive and then negative at first-stage surgery in 7 (5%). No correlation was found between culture results at reimplantation and treatment outcome [18]. In their retrospective study involving 107 patients, Puhto et al. reported 5.2% positive cultures at reimplantation and persistence of infection with the same pathogen in only one case but found no correlation between culture result and treatment outcome [19].

In a 2019 meta-analysis of 10 studies [20], Chi Xu et al. reported 15.2% (141/925 cases) positive cultures at reimplantation, an 18.8% (174/925) total failure rate, and a treatment failure rate of 41.1% (58/141) in patients with a positive culture and 14.8% (116/784) in patients with a negative culture at reimplantation. Analysis of a correlation between treatment failure and persistence of the pathogen isolated at the first stage of revision showed a worse outcome for patients with recurrence of infection with the same pathogen than for those with infection with a different pathogen. The meta-analysis included a retrospective study of 117 patients, 19.7% (23/117) with a positive culture at reimplantation, in 4 (17.4%) of which the pathogen was the same as at explantation. Analysis of the data from the meta-analysis and from the cohort study showed that, despite prolonged antibiotic therapy, the risk of treatment failure was three-fold higher in patients with positive culture at reimplantation than in those with a negative culture. In another study involving 267 cases of PJI (*n* = 186 knee and *n* = 81 hip) treated in two-stage revision, the sample at reimplantation tested positive in 33 (12.4%) patients and for the same pathogen as at explantation in 6/33 (18.2%). Independent of the number of positive samples, the number of positive intraoperative samples was correlated with a two-fold higher risk of treatment failure. [21] This observation is shared by Theil et al. in their very recent retrospective study involving 204 patients and a minimum follow-up of 24 months. The risk of treatment failure was considerably higher in the patients with a single positive sample and in those with multiple positive samples [22]. 

Our data are in line with those reported by Hart [16], Bejon [17] and Puhto [18] and suggest no correlation between positive culture at reimplantation and higher risk of treatment failure. It is good to remind that the tissue samples at the time of reimplantation are collected before performing a further surgical debridement, and therefore a possible positive result is not strictly linked to the clinical evidence of persistence of infection. Furthermore, the patients with a positive culture received prolonged antibiotic therapy, and this may have protected the prosthesis from recurrence of infection.

Analysis of our data also shows that the failure rate was higher in patients with a positive than in those with a negative culture, and the difference was more pronounced for the patients with one positive culture, none of which developed recurrence of infection if categorized as the result of contamination grouped together with group A. Two treatment failures (2.7% and 20%, respectively) were recorded each in the groups with 74 and 10 spacers (negative and positive cultures, respectively). The difference was not statistically significant (*p* = 0.105) but probably might have been in a larger study sample.

We believe that, as suggested by the 2018 Philadelphia Consensus Conference and on the basis of the small patient sample and low number of treatment failures, prolonged antibiotic therapy in patients with a positive culture at reimplantation may protect the prosthesis from recurrence of infection. 

The present study has several limitations. Its retrospective design lends less strength to the evidence than a prospective study and does not allow for determining causal factors but only associations. The small number of patients does not allow definitive conclusions to be drawn.

## 4. Materials and Methods

For this retrospective study, the medical records were reviewed of 126 consecutive patients who underwent two-stage exchange arthroplasty for hip or knee PJI at our institution between May 2018 and September 2020. Inclusion criteria were: PJI diagnosed according to 2013 MSIS criteria; all prosthetic components explanted at stage one, septic tissues debrided; 4 to 6 tissue samples taken from the surgical field and sent to the lab for microbiological testing (bacteria, fungi, microbacteria if specific risk factors were known). 

Microbiological testing was performed according to standard procedures at our institution. Before culture, samples were treated with 0.1% *w:v* dithiothreitol to free the pathogens from the biofilm [23,24]. The eluate obtained after dithiothreitol treatment or sonication was centrifuged and the pellet plated on chocolate agar, MacConkey agar, mannitol salt agar and Sabouraud agar and inoculated onto brain heart infusion or thioglycolate broth. The plates were incubated for 48 h at 37 °C, while the broths were kept at 37 °C for 15 days and checked daily for microbial growth. Aliquots from the brain heart infusion broth showing turbidity were plated on blood agar and those from the thioglycolate broth were plated on Schaedler agar. All colonies grown on agar plates were identified by means of biochemical assays performed on a Vitek2 system (BioMerieux, Marcy L’Etoile, France). Gram positive cocci were identified with a GP card and Gram negative bacilli with a GN card. Cards AST659, AST658 and AST376 were used for antibiotic susceptibility testing of staphylococci, enterococci and Gram negative bacilli. Microbial identification and antimicrobial susceptibility testing were carried out on a Vitek2 system (BioMerieux). 

A fixed or mobile spacer was placed in cement loaded with an antibiotic according to the antibiogram. In the two-stage surgeries, the first stage included implant removal and debridement of any dead tissue. Tissue samples were sent for microbiology (at least 5 specimens, including the removed implant) [25,26].

After saline irrigation, a preformed cement spacer loaded with either gentamycin or vancomycin and gentamycin (*Spacer-G^®^* and *Vancogenx^®^*, respectively, Tecres, Sommacampagna, Italy) was introduced. The choice between the two spacers was based on the available preoperative antibiograms. Alternatively, a custom-made cement spacer, which could be loaded intraoperatively with other antibiotics (meropenem or tobramycin in cases of Gram negative infection resistant to gentamicin, liposomal amphotericin b in the two cases of *Candida albicans* infection) was preferred when vancomycin- and gentamycin-resistant pathogens had been isolated preoperatively. All patients followed a postoperative antibiotic protocol set by our infectious diseases consultant and based on the intraoperative culture antibiogram. 

After explantation, antibiotic therapy was continued for 4 to 6 weeks, at least 2 of which by intravenous administration, then discontinued 4 weeks before reimplantation. Antibiotic therapy was evaluated based on available antibiograms and in agreement with the infection physician. If the germ was unknown, combined vancomycin and carbapenemic therapy was initiated. Reimplantation was timed based on clinical data (signs of local inflammation) and lab results (CRP, ESR), intraoperative leukocyte esterase strip testing of synovial fluid was performed as needed. Reimplantation was performed on average 96 days after first-stage surgery. Debridement was performed again during reimplantation; 4 to 6 tissue samples and the spacer were sent to the lab for microbiological testing. Antibiotic therapy was continued until microbiological tests resulted negative (15 days) or continued for 6 to 8 weeks if one or more cultures tested positive. Antibiotic therapy was evaluated according to the microbiological test results by an infectious disease specialist.

Treatment success or failure was determined according to Delphi criteria [27] at a mean follow-up of 28.5 months (range, 7.0–42.0). Criteria were: eradication of infection and healing of surgical wounds and fistula, if present; no secretions, pain or recurrence of infection caused by the same germ; no further interventions due to infection; no PJI-related mortality. Exclusion criteria were: prosthesis explantation performed at another medical center; reimplantation; reimplantation of spacers; missing clinical data.

### Statistical Analysis

Statistical analysis was performed using R software ver. 4.1.0 (R Core Team, Vienna, Austria). Categorical data are presented as absolute frequencies; differences within subgroups were analyzed with Fisher’s exact test, if applicable, or the chi-squared test. Continuous data distribution was assessed using the Shapiro–Wilk test; the data are reported as median (range) according to non-normal distribution. Differences in these variables between subgroups were analyzed using the Mann–Whitney *U* test. Multiple logistic regression models were used to assess the influence of demographic or clinical variables on the occurrence of new infection. Statistical significance was set at *p* < 0.05. 

## 5. Conclusions

Positive microbial culture at reimplantation raises doubt about whether it increases the risk of treatment failure. Our data indicate a correlation between a positive reimplantation culture and an increased risk of treatment failure. However, we noted a higher rate of treatment failure in patients with at least two positive samples compared to those with one negative culture or one positive culture. Further studies are needed to clarify the implications of this finding.

## Figures and Tables

**Figure 1 antibiotics-11-01408-f001:**
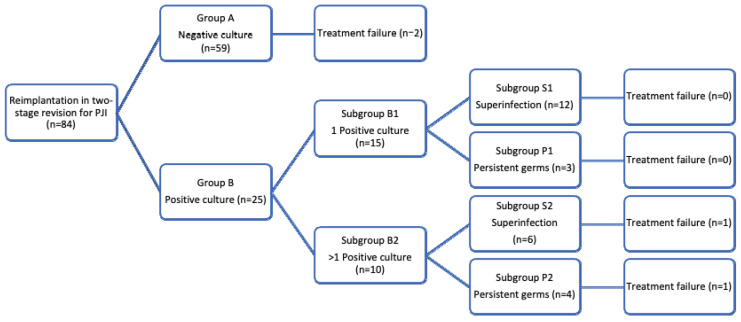
Patients grouped by microbiological test and treatment outcomes.

**Table 1 antibiotics-11-01408-t001:** Patient demographics and clinical characteristics.

	Overall	Treatment Success	Treatment Failure	*p* Value
Age—years (range)	72 (42–85)	71.5 (42–85)	74 (52–80)	0.809
Sex - Men - Women	44 40	41 39	3 1	0.678
Diabetes mellitus - No - Type I - Type II	72 10 2	68 10 2	4 0 0	0.705
BMI	27.7 (17.0–44.0)	27.3 (17.0–44.0)	30.5 (29–33.2)	0.157
Smoker - Yes - No	22 62	21 59	1 3	0.999
Fistula - Yes - No	49 35	48 32	1 3	0.386
Previous surgery - First implant - Aseptic revision - Septic revision	50 12 22	50 12 18	0 0 4	0.003
Follow up—months (range)	28.5 (7.0–42.0)	29.0 (7.0–42.0)	23.0 (14.0–34.0)	0.430
Days with spacer (range)	96 (0–384)	96 (10–384)	95 (0–150)	0.150
Surgery time—min (range)	140 (81–291)	140 (81–291)	146 (117–176)	0.571

Notes: BMI, body mass index (weight in kg divided by height in m^2^); data are expressed as median (range).

**Table 2 antibiotics-11-01408-t002:** Pathogens isolated at explantation and reimplantation.

Microorganism	Explantation (N = 90)	Reimplantation (N = 29)	Total
*Staphylococcus epidermidis*	26	9	35
*Staphylococcus aureus*	20	5	25
*Propionibacterium acnes*	4	4	8
*Corynebacterium striatum*	5	2	7
*Enterococcus faecalis*	5	1	6
*Staphylococcus capitis*	4	2	6
*Staphylococcus hominis*	2	2	4
*Staphylococcus warneri*	4	0	4
*Pseudomonas aeruginosa*	4	0	4
*Staphylococcus lugdunensis*	3	0	3
*Staphylococcus haemolyticus*	2	0	2
*Candida albicans*	2	0	2
*Serratia marcescens*	2	0	2
*Klebsiella pneumoniae*	1	1	2
*Staphylococcus lentus*	1	0	1
*Staphylococcus caprae*	0	1	1
*Escherichia coli*	0	1	1
*Streptococcus pyogenes*	1	0	1
*Staphylococcus saprophyticus*	0	1	1
*Citrobacter koseri*	1	0	1
*Brevibacterium*	1	0	1
*Prevotella bivia*	1	0	1
*Streptococcus sanguis*	1	0	1

## Data Availability

Not applicable.

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
