# Peer review of "Reimplantation after Periprosthetic Joint Infection: The Role of Microbiology"

_antibiotics, 2022, doi:10.3390/antibiotics11101408_

Round 1

Reviewer 1 Report (Previous Reviewer 1)

I am satisfied with the vast majority of the changes made by the authors. I only disagree with one thing: in the conclusions, the authors state "However, we motivated a higher, albeit not statistically significant, rate of treatment failure in patients with at least two positive samples compared to those with one negative culture or one positive".  If there is no significant difference, it should not be mentioned because it devalues the statistical study performed. I recommend that this sentence be completely removed from the manuscript before publication.

Author Response

Dear Reviewer thank you for your comment and your evaluation of our paper.

Unfortunately we noted that you reported a changed version of our conclusion, actually we wrote as follow: 

"Positive microbial culture at reimplantation raises doubt about whether it increases the risk of treatment failure. Our data indicate correlation between a positive reimplantation culture and an increased risk of treatment failure. However, we noted a higher, albeit not statistically significant, rate of treatment failure in patients with at least two positive samples compared to those with one negative culture or one positive culture. Further studies are needed to clarify the implications of this finding."

If you agree we could change the sentence deleting "albeit not statistically significant", in order not to make our conclusion weak.

Best Regards

Reviewer 2 Report (New Reviewer)

The article apparently has been mixed up (results and methods part) during the revision process and needs to be overworked in a correct sequential order before continuing with the review process. At present, the reviewer is forced to gather the respective information from different parts of the article, which is annoying and impedes reading.

Some remarks with regard to necessary amendments changes have been added:

Table 1 please explain /modify previous surgery none, otherwise might be misleading (e.g., no previous revision surgery?)

Table 2: Please check numbers, it is 90 and 29, respectively.

Line 161: persistence instead of persistent?

Line 219 typo interoperative (should run intraoperative?)

Lines 202-203: please give further details with regard to added antibiotics: there is just a limited spectrum of antibiotics for cements available (vanco and genta), so what to do if the antibiogram doesn’t fit? Never happened? 

Please check numbers in the abstract (84, all revisions? All 2-stage?), as at present the running text is quite confusing due to the mix-up.

Last but not least: please explain already in the introduction what is the new research question and in the discussion with regard to the new  information achieved by your results in contrast to the larger series of Bejon et al (152) and Putho et al.(107) to make clear that your article is not a me too.

In addition, please also correct my reviewer statistics, many thanks (not overdue)

Author Response

Dear Reviewer thank you for your valuable suggestions.
Below are the point-by-point answers to your inquiries.

The article apparently has been mixed up (results and methods part) during the revision process and needs to be overworked in a correct sequential order before continuing with the review process. At present, the reviewer is forced to gather the respective information from different parts of the article, which is annoying and impedes reading.

Dear Reviewer, thank you for the suggestion. We have edited the paper by moving the results to the correct paragraph. We hope it will be easier to read this way.

Some remarks with regard to necessary amendments changes have been added:

Table 1 please explain /modify previous surgery none, otherwise might be misleading (e.g., no previous revision surgery?) 

We modified the table adding previous surgery: first implant 50 patients, aseptic revision 12 patients, septic revision 22 patients

Table 2: Please check numbers, it is 90 and 29, respectively.

Thank you we corrected it.

Line 161: persistence instead of persistent?

Thank you we corrected it.

Line 219 typo interoperative (should run intraoperative?)

Thank you we corrected it.

Lines 202-203: please give further details with regard to added antibiotics: there is just a limited spectrum of antibiotics for cements available (vanco and genta), so what to do if the antibiogram doesn’t fit? Never happened? 

Thank you for your comment but we explain your question from line 206 to line 211 , nevertheless now we specified the antibiotics added in the custom made spacers as follow: "Alternatively, a custom-made cement spacer, which could be loaded intraoperatively with other antibiotics (meropenem or tobramycin in cases of gram-negative infection resistant to gentamicin, liposomal amphotericin b in the two cases of Candida albicans infection) was preferred when vancomycin- and gentamycin-resistant pathogens had been isolated preoperatively"

Please check numbers in the abstract (84, all revisions? All 2-stage?), as at present the running text is quite confusing due to the mix-up.

Dear Reviewer thank you for your comment. The aim of our study is to check the role microbiology after two stage revision and therefore the 84 patients underwent a two stage revision procedure.

Dear Reviewer, we apologize but we did not understand what is required in the following points:

Last but not least: please explain already in the introduction what is the new research question and in the discussion with regard to the new  information achieved by your results in contrast to the larger series of Bejon et al (152) and Putho et al.(107) to make clear that your article is not a me too. Sorry but what is meant by "is not a me too"?

In addition, please also correct my reviewer statistics, many thanks (not overdue)

Sorry but what is meant by "correct my reviewer statistics"?

Thank you, best regards

Round 2

Reviewer 2 Report (New Reviewer)

Thanks a lot for performing the requested changes, the article is now much better readable

Typos just as noted: please check

line 135: guess this should run test 

line 185: remind or kept in mind instead of remember?

line 253: please eliminate to in results. by

FYI:  a "me too" is an article, which does not have ab´n original research question, butte-investigates what other have already done. Therefore, an rattle should strive at clarifying which is the new new approach/new information given in the article. Lines 70-71 state the goal of the article, but it remains unclear, whether this is a novel approach. In case of the present article it becomes clear in the discussion part that there is a controversy 148-149.

Author Response

Thank you very much for your comment.

We have corrected as follow:

- line 135: guess this should run test 

Thank you, we corrected it

- line 185: remind or kept in mind instead of remember?

Thank you, we corrected it

- line 253: please eliminate to in results. by

Thank you, we corrected it

- FYI:  a "me too" is an article, which does not have ab´n original research question, butte-investigates what other have already done. Therefore, an rattle should strive at clarifying which is the new new approach/new information given in the article. Lines 70-71 state the goal of the article, but it remains unclear, whether this is a novel approach. In case of the present article it becomes clear in the discussion part that there is a controversy 148-149.

Thank you very much for the explanation, we changed the aim in the introduction part as follow to make it clearer:

"The aim of the present study is to report our experience about the hypothetical correlation between culture results at reimplantation after spacer insertion for treating hip and knee PJI and treatment failure rate"

This manuscript is a resubmission of an earlier submission. The following is a list of the peer review reports and author responses from that submission.

Round 1

Reviewer 1 Report

I am satisfied with the changes made by the authors and the justifications provided.

Reviewer 2 Report

This study aims to investigate the impact of microbiological testing at reimplantation however, fails to describes the individual adjustments based on the findings. 

Typically, if a persisting PJI is suspected, a biopsy or aspiration should be obtained couple of weeks prior to reimplantation to evaluate if it is safe to do so. Alternatively, a suppression antibiotic therapy needs to be considered depending on the pathogen found. 

The study group did not differentiate between THA and TKAs which is crucial to describe the outcome. In addition, the actual antibiotic recommendations are not clearly described. The type of initial implant is also important including cemented or non cemented implant.

I don't believe this article adds much to the current knowledge on PJI.

Line 18: exists

Please be uniform: PJI or arthroplasty infection.

Line 42: thats wrong. It depends on the bacteria and type of PJI - chronic vs. acute. In acute PJIs single stage or simple wash outs can be performed.

Line 63: There are guidelines, although no real gold standard which surgeons can follow.

If microbiological testing is performed at reimplantation surgeons don’t really have a choice to wash out without reimplantation.

Did you perform a hip aspiration. What was the antibiotic treatment, how long for? Was a suppressive therapy performed after reimplantation?

Why is materials and methods section 4??

Did you obtain a sonication?

When did you use a fixed when a mobile spacer? Why not a Girdlestone situation?

Reviewer 3 Report

In the presented manuscript, the authors investigate outcome in patients with PJI of the hip and knee that present themselves with positive microbiological culture at prosthesis reimplantation. While the topic is of great clinical relevance, the manuscript analyzes too few patients to draw valid conclusions from. Additionally, the manuscript lacks novelty as extensive research on exactly the same topic has previously been done by other groups (e.g. doi: 10.2106/JBJS.15.01469).

If the authors want to be reconsidered for a resubmission the following points should be addressed:

  1. Please fix a lot of the punctuation mistakes throughout the text and abstract
  2. Include more patients and investigate longer-follow up times to be able to make a sound conclusion
  3. Explain the novelty of this research in context of previously published work